# Membrane translocation process revealed by in situ structures of type II secretion system secretins

Zhili Yu[1,6], Yaoming Wu[2,6], Muyuan Chen[1,5], Tong Huo[1], Wei Zheng[2], Steven J. Ludtke [1,3], Xiaodong Shi[2] & Zhao Wang [1,3,4]

The GspD secretin is the outer membrane channel of the bacterial type II secretion system (T2SS) which secrets diverse toxins that cause severe diseases such as diarrhea and cholera. GspD needs to translocate from the inner to the outer membrane to exert its function, and this process is an essential step for T2SS to assemble. Here, we investigate two types of secretins discovered so far in *Escherichia coli*, $GspD_\alpha$, and $GspD_\beta$. By electron cryotomography subtomogram averaging, we determine in situ structures of key intermediate states of $GspD_\alpha$ and $GspD_\beta$ in the translocation process, with resolution ranging from 9 Å to 19 Å. In our results, $GspD_\alpha$ and $GspD_\beta$ present entirely different membrane interaction patterns and ways of transitioning the peptidoglycan layer. From this, we hypothesize two distinct models for the membrane translocation of $GspD_\alpha$ and $GspD_\beta$, providing a comprehensive perspective on the inner to outer membrane biogenesis of T2SS secretins.

Secretion systems, which are bacterial cell envelope-located protein complexes, are utilized by bacteria to produce virulence-related substrates that facilitate survival and pathogenicity[1]. Among secretion systems, the type II secretion system (T2SS) is broadly present and functional in *Proteobacteria* species, including non-pathogenic *Escherichia coli* (*E. coli*), Enterotoxigenic *E. coli* (ETEC), Enteropathogenic *E. coli* (EPEC), *Vibrio cholerae, Klebsiella pneumoniae*, and *Aeromonas hydrophila*[2]. T2SS substrates in non-pathogenic bacteria can facilitate nutrient absorption from the environment or symbiosis with plants or animals, whereas in pathogenic bacteria, T2SS substrates can aid in adhesion to hosts, intoxicate host cells, and suppress immunity in the host, causing various diseases[3]. With T2SS's diverse substrate functions and close relevance to virulence and diseases, knowing its structure and working mechanism is necessary for understanding bacteria functions and developing antimicrobial strategies.

The outer membrane component of T2SS is the secretin, constituting a large channel structure, connecting to the protein scaffold on the inner membrane, and controlling the last step of substrate transportation[2]. Phylogenetic analysis of T2SS secretins from *Proteobacteria* species demonstrated two types: the *Klebsiella*-type secretins found in *Klebsiella* and *Dickeya*; and *Vibrio*-type secretins found in *Vibrio*, ETEC, and EPEC[4]. Structures of the two secretin types both appear as cylindrical channels containing the N0-N3 domains, the secretin domain with a central gate region, and the S domain[5–10]. But they are different in the transmembrane region: the *Vibrio*-type secretins have about 20 additional amino acids between the α7 and α8 helices, forming a loop extending upward and inward to the central axis of the channel like a roof, constituting a cap gate, while *Klebsiella*-type secretins do not have the cap gate[5–10]. This directly causes differences in the height of the proposed transmembrane regions of the two secretin types, which may give them different ways of interacting

[1]Verna and Marrs McLean Department of Biochemistry and Molecular Biology, Baylor College of Medicine, Houston, TX 77030, USA. [2]Jiangsu Province Key Laboratory of Anesthesiology and Jiangsu Province Key Laboratory of Anesthesia and Analgesia Application, Xuzhou Medical University, Xuzhou, Jiangsu 221004, China. [3]Cryo Electron Microscopy and Tomography Core, Baylor College of Medicine, Houston, TX 77030, USA. [4]Department of Molecular and Cellular Biology, Baylor College of Medicine, Houston, TX 77030, USA. [5]Present address: Division of CryoEM and Bioimaging, SSRL, SLAC National Accelerator Laboratory, Stanford University, Menlo Park, CA 94025, USA. [6]These authors contributed equally: Zhili Yu, Yaoming Wu. ✉e-mail: shixd@xzhmu.edu.cn; zhaow@bcm.edu

with the membrane. The in situ structure of T2SS has been visualized in *Legionella pneumophila*, which has indicated the architecture of T2SS and the membrane interaction of the secretin[11]. However, the resolution is limited to a few nanometers, hindering the observation of more details from the structure. Studying high-resolution structures of secretins in situ could reveal possible biological processes that occur in the cellular environment and offer a more complete understanding of their assembly process and mechanisms. This environment cannot be accurately simulated in vitro and may have a significant impact on secretin behavior.

The translocation of T2SS secretins from the inner membrane to the outer membrane is an essential step required for T2SS assembly, however, it is still unclear how this translocation is regulated in the bacterial cell envelope. Based on biochemistry experiments, there exist two possible pathways: (1) the scaffolding proteins GspA and GspB bind and increase the pore size of peptidoglycan, and translocate the secretin to the outer membrane, as found in *Klebsiella*-type secretins ExeD and OutD[12–14]; (2) a small pilotin GspS could bind to the S domain of secretin, and the pilotin itself can translocate to the outer membrane through the Lol sorting pathway, as found in the *Vibrio*-type secretins[4,15,16]. Besides these two pathways, since some T2SS exist without GspA and GspB or the pilotin[11,17,18], other pathways also possibly exist. Despite the current biochemical evidence, this translocation process has not been visualized in living cells. By doing in situ electron cryotomography (cryo-ET) studies on T2SS secretins, we capture different intermediate states during the outer membrane translocation process, which helps to explain the translocation mechanism and provide a clearer blueprint of the biogenesis process of T2SS secretins.

There are two T2SS secretins encoded in two T2SS operons in the *E. coli* genome: the *Klebsiella*-type $GspD_{\alpha}$ encoded in the $T2SS_{\alpha}$ operon that can secrete chitinase[19–22], representing non-pathogenic functions; and the *Vibrio*-type $GspD_{\beta}$ encoded in the $T2SS_{\beta}$ operon[4,15], which can secrete toxins[23,24], representing pathogenic functions. In this study, we investigate the in situ structures of $GspD_{\alpha}$ from the *E. coli* K12 strain and $GspD_{\beta}$ from the ETEC H10407 strain as representatives of the *Klebsiella*-type secretins and *Vibrio*-type secretins, respectively. We report four in situ structural states of specific secretin/secretin-pilotin complexes, determined within *E. coli* cells, using the cryo-ET subtomogram averaging. They are, respectively, $GspD_{\alpha}$ on the inner membrane, $GspD_{\alpha}$ on the outer membrane, $GspD_{\beta}$–GspS complex on the outer membrane, and $GspD_{\beta}$ on the inner membrane (Supplementary Table 1). Together, these structures show interactions of secretins with inner and outer membranes and provide insights into the biogenesis process of the GspD secretin.

## Results

### Visualization of $GspD_{\alpha}$ on the inner membrane through cryo-ET
To obtain thin cells for improved contrast under cryo-ET, we employed a bacteria minicell system to provide thin cells which retain physiological activities[25]. We induced overexpression of $GspD_{\alpha}$ within *E. coli* BL21 (DE3) cells (Fig. 1j). Cryo-ET was performed to image the minicells and 250 tilt series were collected. From the raw tilt images and the reconstructed tomograms, cell features including the intact bacterial cell envelope and membrane-bound $GspD_{\alpha}$ particles could be clearly recognized (Fig. 1a–d). $GspD_{\alpha}$ multimer particles in different orientations with respect to the grid plane could be identified in the tomograms, and surprisingly, under these experimental conditions, the $GspD_{\alpha}$ particles are naturally located on the inner membrane of the bacterial envelope, with the non-transmembrane domains facing the periplasmic space (Fig. 1d). As $GspD_{\alpha}$ has an approximately cylindrical shape, we identified particles in the top view and side view as in circles and a pair of curved lines attaching to the inner membrane, respectively. The images of negative control cells

further validated that these particle features found in the tomograms belong to our protein of interest (Supplementary Fig. 1).

### In situ structure of $GspD_{\alpha}$ at subnanometer resolution
For subtomogram averaging, ~32,000 particles were manually picked, and data processing was carried out using EMAN2[26]. In a series of previous in vitro structures, it has been verified that secretin has C15 symmetry[5,7,9,27,28]. However, the symmetry of secretin in vivo has not been verified. To confirm the symmetry of our particles, we developed an algorithm to measure the radius of individual particles and generated a histogram to look for potential multiple radii (Supplementary Fig. 2). The histogram shows only one peak with a diameter consistent with the C15 structure of $GspD_{\alpha}$ (PDB: 5WQ7), strongly implying that there is only one diameter of $GspD_{\alpha}$ particles in situ on the inner membrane, corresponding to its C15 structure. One challenge in subtomogram alignment of the secretin is that, due to the lack of low-resolution features along the symmetry axis, at the initial stage of orientation determination, it is difficult to distinguish top-view particles from bottom-view ones without information about the membrane position. Therefore, at the beginning of the subtomogram refinement, many top-view particles were misaligned and positioned upside down, limiting the final resolution, and causing the initial reconstructed map to exhibit some "D" symmetric features (symmetry along the central *x–y* plane) in addition to the 15-fold symmetry. To resolve this ambiguity, we developed an algorithm in the EMAN2 tomography package, which considers the geometric location of the cell membrane as an additional factor in particle orientation determination (see "Methods"). The use of this methodology improved both visible features in the map as well as the measured resolution (Supplementary Fig. 3b, c).

The final C15 symmetrized subtomogram average achieved a measured resolution of 9 Å (Supplementary Fig. 4a), demonstrated by some visibility of alpha helices in the map (Supplementary Fig. 5b, Supplementary Movie 1). The inner membrane bilayer is clearly resolved and visible at the bottom of this density map. Above the inner membrane bilayer are the spool-shaped protein (cylinder with a larger diameter at both ends) with a sealed membrane adjacent surface and an open membrane distal surface. The N0–N3 domains can be clearly recognized in the density map (Supplementary Fig. 5b). In N1 and N2 domains, the two layers of alpha-helices are just resolved, consistent with the measured resolution of the map. With the position of the N domains identified, the transmembrane region of $GspD_{\alpha}$ can be approximately modeled using the length of the secretin domain. The tip of $GspD_{\alpha}$ (α7–β11, β14, and the membrane-buried region of β10 and β15[5]) transits through one leaflet of the lipid bilayer but does not penetrate through or generate an opening on the membrane (Supplementary Fig. 5b). The sealed end of the cylinder corresponds to the gate region of the secretin. The connecting density between the protein and the inner membrane appears to have very low occupancy based on the low isosurface threshold required to visualize it. Interpreting this connection thus required some additional effort.

### $GspD_{\alpha}$ is flexible on the bacterial inner membrane
As both the membrane and $GspD_{\alpha}$ particles are clearly resolved in the raw tomogram, we could directly observe that some particles were slightly tilted with respect to the membrane, instead of perpendicular to the membrane (Fig. 1e–g). To further confirm the membrane connection of $GspD_{\alpha}$, using the C15 density map and the corresponding orientation information of each particle, we generated an asymmetric structure by relaxing the symmetry (Supplementary Fig. 3e, f) (see "Methods"), producing a structure with a reduced resolution of 15 Å (Supplementary Fig. 5d). In this density map, the inner membrane leaflets are still resolved, as expected, and $GspD_{\alpha}$ constitutes the

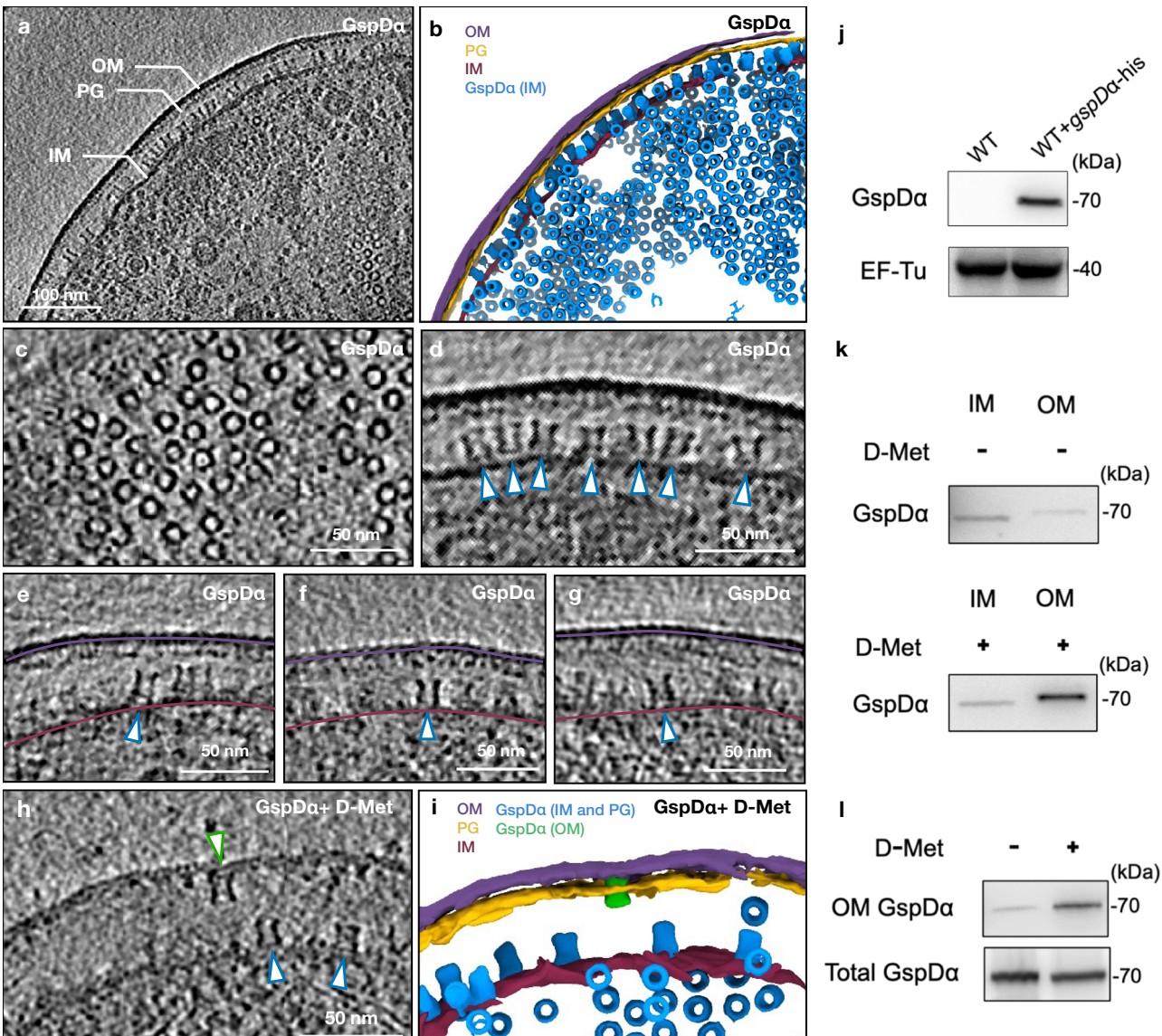

**Fig. 1 | Visualization of GspD$_\alpha$ multimers within *E. coli* cells and determination of the overexpression and membrane location of GspD$_\alpha$. a** The tomogram z slice view of *E. coli* overexpressing GspD$_\alpha$. **b** Segmentation of the tomogram shown in (**a**), with OM colored purple, PG colored yellow, IM colored dark red, and GspD$_\alpha$ on IM colored blue. **c** Zoom in tomogram z slice view of top view GspD$_\alpha$ particles. **d** Zoom in tomogram z slice view of side view GspD$_\alpha$ particles, indicated by white arrowheads with blue outline. **e**–**g** Tomogram z slices showing tilted GspD$_\alpha$ on the inner membrane or GspD$_\alpha$ with only one side connecting the inner membrane (white arrowheads with blue outline). The outer and inner membranes are indicated by the purple line and the dark red line, respectively. **h** A tomogram z slice view of *E. coli* overexpressing GspD$_\alpha$ and with D-methionine added (the GspD$_\alpha$ particle on the OM is indicated by a white arrowhead with a green outline). **i** Segmentation of the tomogram shown in h, with OM colored purple, PG colored

yellow, IM colored dark red, GspD$_\alpha$ on IM colored blue, and GspD$_\alpha$ on OM colored green. **j** Immunoblotting results of GspD$_\alpha$ overexpression by using anti-His tag antibodies. BL21 (DE3) cells (WT) without plasmid transformation were used as the control. k, Inner and outer membrane fractions of the GspD$_\alpha$ overexpressing cells treated with/without D-methionine were separated by sucrose density gradient centrifugation. The membrane location of GspD$_\alpha$ was immunoblotted with anti-His tag antibodies. **l** Outer membrane proteins of the GspD$_\alpha$ overexpressing cells treated with/without D-methionine were extracted using a bacterial membrane protein extraction kit. The total GspD$_\alpha$ and outer membrane GspD$_\alpha$ were examined by western blot analysis using anti-His tag antibodies. The expression and membrane location experiments of GspD$_\alpha$ were repeated three times independently with similar results. IM inner membrane, OM outer membrane, PG peptidoglycan, D-Met D-methionine, WT wild type. Source data are provided as a Source Data file.

middle-contracted cylindrical density. The membrane connecting density is now clearly visible with apparent full occupancy, but only on one side of the cylinder. The membrane connection covers a roughly 100° arc, depending on the threshold level, so when bound to the inner membrane, GspD$_\alpha$ is only partially connected. When C15 symmetry is applied along the cylindrical axis, this partial connection is averaged out, yielding the lower apparent occupancy in the original symmetrized structure (Supplementary Fig. 5b).

In addition, we performed a focused refinement on the protein region, excluding the membrane, and compared the particle

orientation with that of the integrated refinement result (Supplementary Fig. 3g) (see "Methods"). The orientation difference between the two refinements can then be determined for each particle. By classifying these differences, we recovered a trajectory for GspD$_\alpha$ on the inner membrane, where the multimer swings around the membrane contact site (Supplementary Movie 2). We show the two endpoints' conformations for this trajectory (Fig. 2b,c): in conformation A, the symmetry axis of GspD$_\alpha$ is perpendicular to the membrane; and in conformation B, the symmetry axis of GspD$_\alpha$ is tilted 2.86° compared to conformation A.

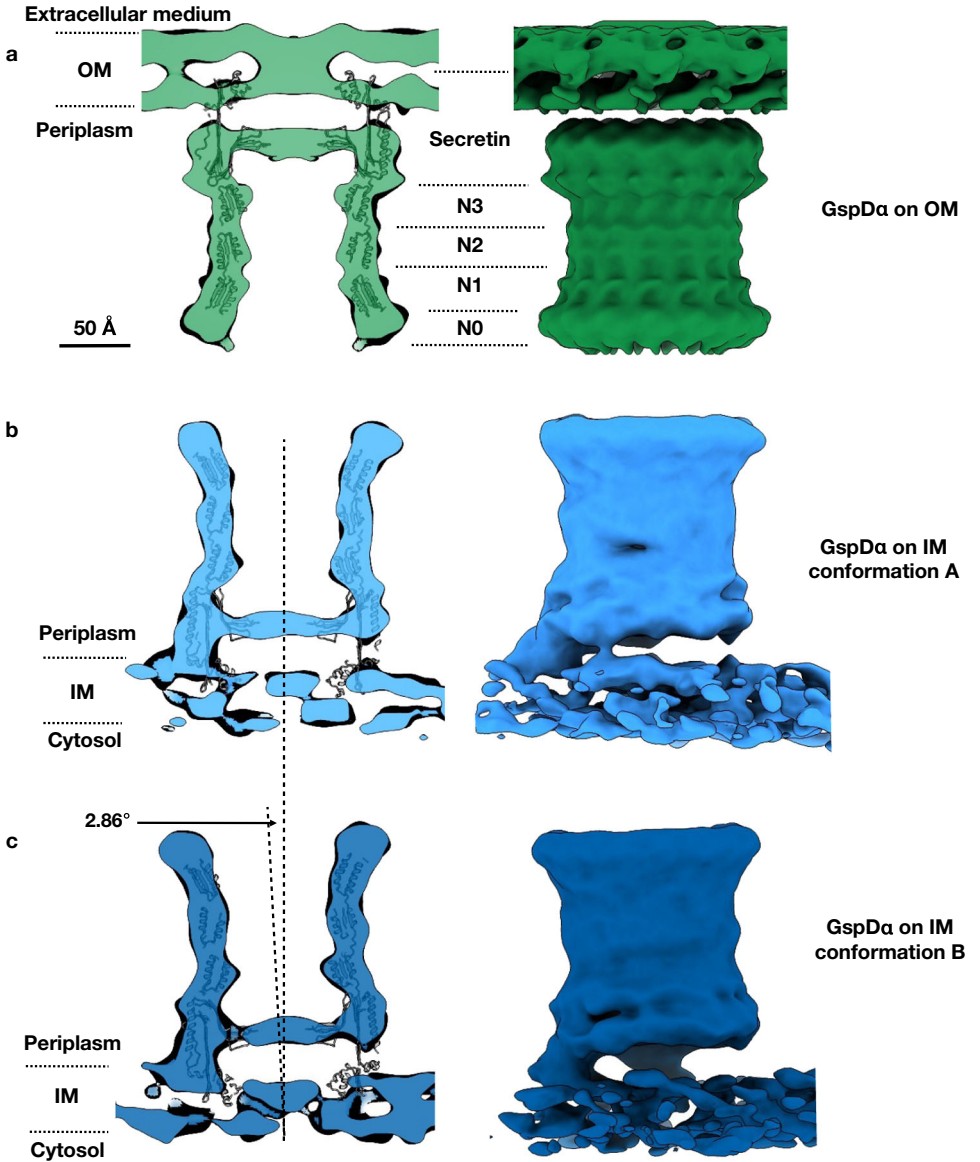

**Fig. 2 | In situ structures of the GspD$_\alpha$ secretin on the outer and inner membranes. a** The in situ structure of GspD$_\alpha$ on the outer membrane, showing the central slice view and side view. The secretin, N3, N2, N1, and N0 domains are marked with dashed lines. **b**, **c** The in situ structures of GspD$_\alpha$ conformation A and conformation B on the inner membrane respectively, showing the central slice view and side view. Density maps are fitted by the GspD$_\alpha$ in vitro structure (PDB: 5WQ7). OM outer membrane, IM inner membrane.

In summary, these results indicate that the GspD$_\alpha$ multimer is not integrated into the bacterial inner membrane. Instead, it only forms a loose connection with significant mobility, facilitating its further transportation to the outer membrane (see "Discussion").

## GspD$_\alpha$ structure on the bacterial outer membrane

We hypothesize that the GspD$_\alpha$ multimer is on the inner membrane because of the existence of the peptidoglycan, which is a meshwork whose pore size is too small for the GspD$_\alpha$ multimer to pass through and translocate to the outer membrane. We attempted to reduce peptidoglycan crosslinking to see if this would facilitate GspD$_\alpha$ outer membrane targeting. In previous studies, it was shown in *Aeromonas hydrophila* that decreasing the cross-linking of the peptidoglycan by glycine could localize the secretin ExeD to the outer membrane[12,13]. Biochemical evidence also showed that the growth of *E. coli* in media containing a high concentration of D-methionine decreases the cross-linking of the *E. coli* peptidoglycan[29], the mechanism of which includes both D-methionine incorporating into the muropeptides and D-

methionine directly influencing peptidoglycan remodeling. Therefore, we raised *E. coli* minicells in LB medium containing 40 mM D-methionine, induced expression of GspD$_\alpha$, and found that this increased the proportion of GspD$_\alpha$ on the outer membrane compared to the control (Fig. 1k, l). Within the tomograms, we observed GspD$_\alpha$ particles located on the outer membrane as well as the inner membrane (Fig. 1h, i). For subtomogram averaging, we only selected particles on the outer membrane. There were 309 particles, leading to a final refinement with C15 symmetry at 16 Å resolution (Supplementary Fig. 4b). Within the density map, the outer membrane bilayer is again resolved with an adjacent spool-shaped density. The membrane adjacent surface of the cylinder is sealed, indicating the gate region. The N0, N1, N2, and N3 domains could be recognized from the membrane distal to the membrane adjacent side of the cylinder (Fig. 2a).

To further investigate the protein-membrane interactions, we relaxed the C15 symmetry as previously for GspD$_\alpha$ on the inner membrane. The resulting density map, seen from the cross-section at the membrane connection, shows GspD$_\alpha$ connecting to the membrane

with an even distribution of occupancy on the contour of the cylinder periphery (Supplementary Fig. 5c), instead of the one-side connection observed on the inner membrane (Supplementary Fig. 5d). To examine the symmetry of $GspD_\alpha$ on the outer membrane, we did a refinement imposing only C5 symmetry if the structure has C15 symmetry, it should show 3 structural repeats within one C5 symmetry unit. The resulting density map retains clear C15 symmetrical features (Supplementary Fig. 5e), confirming the symmetry of $GspD_\alpha$ in situ. For comparison, we did C5 refinement on the $GspD_\alpha$ on the inner membrane dataset, and the resulting density map did not manifest clear C15 features (Supplementary Fig. 5f). Together, these results show that enlarging the pore size of peptidoglycan by adding D-methionine when raising *E. coli* permits $GspD_\alpha$ to translocate to the outer membrane, where $GspD_\alpha$ adopts a more consistent conformation and forms an evenly-distributed connection with the membrane.

### $GspD_\beta$ on the bacterial outer membrane co-exists with GspS

$GspD_\beta$, as a homolog of $GspD_\alpha$, has a similar structure except for an additional cap gate on the membrane adjacent side[5,8], which may generate different transmembrane regions and membrane interactions. Also, $GspD_\beta$ has a different outer membrane targeting mechanism. Instead of using scaffolding proteins or enlarging the peptidoglycan pore size, a small protein GspS, named pilotin, could bind to the S domain of $GspD_\beta$ and translocate $GspD_\beta$ to the outer membrane[4,15]. Therefore, to visualize $GspD_\beta$ particles on the outer membrane and verify whether GspS forms a complex with $GspD_\beta$ there, we performed two experiments. One is that the $GspD_\beta$ and GspS expression are both induced in wild-type *E. coli* minicells, the other one is that only $GspD_\beta$ expression is induced in wild-type *E. coli* minicells, and the protein expression was verified (Fig. 3h).

In the reconstructed tomograms of $GspD_\beta$–GspS expressed cells, particles are all found on the outer membrane (Fig. 3a), verified by a membrane separation experiment (Fig. 3i). We boxed 514 subtomogram particles, and the refinement with C15 symmetry achieved a 19 Å resolution (Supplementary Fig. 4c). To verify the symmetry, we did a refinement with C5 symmetry applied, and the resulting density map still shows C15 features (Supplementary Fig. 6a), confirming that $GspD_\beta$ in situ has C15 symmetry. In the density map (Fig. 3b), the outer membrane density is visible as the top layer. Below the outer membrane is the spool-shaped density, similar to that of $GspD_\alpha$. The membrane adjacent surface of the cylinder density is sealed, indicating the gate region. Notably, extra density appearing as 15 lumps can be seen connecting to the membrane adjacent periphery of the cylinder. These extra densities likely belong to GspS, indicating that $GspD_\beta$ on the outer membrane exists in a complex with GspS. The in vitro structure of the $GspD_\beta$–GspS complex (PDB: 5ZDH) could be well fitted into the density map, and according to the position of the membrane density, the transmembrane region could be located at α7–α10, β11–β14, and the membrane buried region of β10 and β15[5,8]. In contrast to $GspD_\alpha$ which only transits through one leaflet of the membrane, $GspD_\beta$ transit through two leaflets of the lipid bilayer, with the additional cap gate contacting the outer leaflet of the outer membrane.

We expect that in cells where only $GspD_\beta$ is expressed, particles will be on the outer membrane since there should be the endogenous expression of GspS in the *E. coli* BL21(DE3) strain. Observed from the tomograms, most particles are on the outer membrane (Fig. 3c), but surprisingly, a few particle side views are seen on the inner membrane (Fig. 3d), which was verified with a membrane separation experiment (Fig. 3i). Within this dataset, 910 particles were picked, including a mixture of outer and inner membrane-located particles. After one round of refinement, we performed a multi-reference classification with two references rotated 180° with respect to each other, which produced two populations of particles with a ratio of 723 outer membrane particles to 187 inner membrane particles. We used the 723

particles to do the refinement with C15 symmetry and achieved a 16 Å resolution structure (Fig. 3e and Supplementary Fig. 4d). Refinement with C5 symmetry was also performed, and the density map showed C15 features (Supplementary Fig. 6b). This structure basically resembles that of the $GspD_\beta$-GspS expressed dataset (Fig. 3b), but notably, the GspS density is still seen outside the $GspD_\beta$ channel, appearing as lumps connecting to the cylinder surface, although we did not induce overexpression of GspS.

Together, these results indicate that, when GspS is overexpressed together with $GspD_\beta$, they form a complex on the outer membrane. Without exogenous GspS expression, when $GspD_\beta$ is expressed, the *E. coli* endogenous GspS is sufficient to locate $GspD_\beta$ to the outer membrane, where GspS stays attached to $GspD_\beta$.

### $GspD_\beta$ multimer is visualized on the bacterial inner membrane

As biochemistry results have shown that $GspD_\beta$ could form multimers on the inner membrane when *gspS* is knocked out[4,15], to visualize $GspD_\beta$ on the inner membrane, we induced expression of $GspD_\beta$ within Δ*gspS E. coli* cells (Fig. 3h). Within the reconstructed tomograms, particles are all visualized on the inner membrane, with the non-transmembrane domains located in the periplasm (Fig. 3f, i). There are 370 particles, and the refinement with C15 symmetry achieved a 14 Å resolution structure (Supplementary Fig. 4e), rendering $GspD_\beta$ on the inner membrane (Fig. 3g). The membrane density could be located at the bottom, and the cylinder density appears above, inserted into the membrane. The sealing at the lower end of the cylinder corresponds to the gate region. The lower end of $GspD_\beta$ transits through two leaflets of the inner membrane, and there is no GspS density seen outside the cylinder density. These results indicate that, when *gspS* is knocked out, $GspD_\beta$ could self-assemble into a multimer that has stable conformation on the inner membrane.

## Discussion

In this study, we show the in situ structures of the $GspD_\alpha$ secretin from bacterial T2SS and their interactions with the inner and outer membranes. We show that: (1) $GspD_\alpha$ could self-assemble into its multimeric form on the inner membrane without overexpression of any other protein (Fig. 1d); (2) $GspD_\alpha$ multimer forms flexible connections with the inner membrane, and does not generate an opening on the inner membrane (Fig. 2b, c); (3) the undisturbed normal peptidoglycan pore size does not allow the passage of a $GspD_\alpha$ multimer[30]; (4) dissociative $GspD_\alpha$ multimers exist in the periplasm of *E. coli* expressing $GspD_\alpha$, but only between the peptidoglycan and the inner membrane (Supplementary Fig. 7a–c); (5) D-methionine could reduce peptidoglycan crosslinking[29], and adding D-methionine when raising *E. coli* could localize $GspD_\alpha$ to the outer membrane (Fig. 1h); (6) dissociative $GspD_\alpha$ multimers exist in the periplasm of *E. coli* raised with D-methionine and expressing $GspD_\alpha$ (Supplementary Fig. 7d); (7) $GspD_\alpha$ on the outer membrane exists in a more stable conformation, showing distinguishable symmetry, and forming evenly-distributed connections with the outer membrane (Supplementary Fig. 5c, e). Altogether, this evidence supports a translocation model in which $GspD_\alpha$ first forms multimers on the inner membrane, where it stays in an intermediate state, unstable and swinging, which could favor its transportation to the outer membrane. When the peptidoglycan pore is enlarged by D-methionine, the multimer translocates to the outer membrane without disassembly. On the outer membrane, it exists in a consistent conformation, firmly attached to the membrane, which potentially facilitates substrate transportation (Fig. 4a). Notably, in our data, after we treat *E. coli* cells with D-methionine, not all $GspD_\alpha$ particles are located on the outer membrane. This result indicates that other pathways or factors possibly exist to help $GspD_\alpha$ translocate to the outer membrane, such as the GspA and GspB proteins encoded in the $T2SS_\alpha$ operon. We acknowledge that the inner membrane-associated state results are observed from overexpression bacteria and the states we

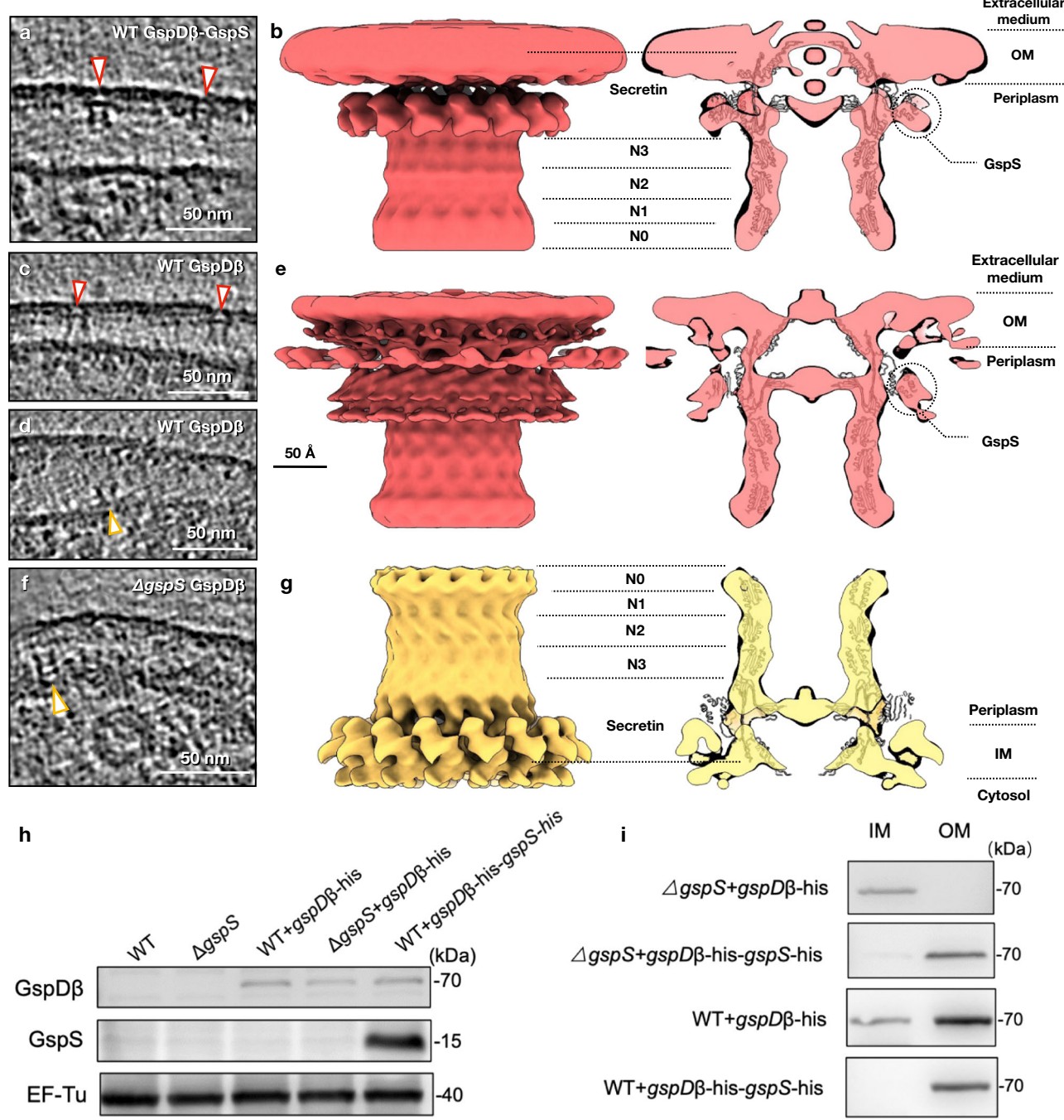

**Fig. 3 | In situ structures of the GspD_β secretin and determination of the overexpression and membrane location of GspD_β.** **a** The tomogram z slice view of *E. coli* overexpressing GspD_β and GspS. **b** The in situ structure of the GspD_β–GspS complex on the outer membrane when both GspD_β and GspS are overexpressed, showing the side view and central slice view. **c**, **d** Tomogram z slice views of *E. coli* overexpressing only GspD_β. **e** The in situ structure of the GspD_β–GspS complex on the outer membrane when only GspD_β is overexpressed, showing the side view and central slice view. **f** The tomogram z slice view of *ΔgspS E. coli* overexpressing GspD_β. **g** The in situ structure of the GspD_β multimer on the inner membrane, showing the side view and central slice view. Particles on the outer and inner membranes are indicated by white arrowheads with a red outline and white arrowheads with a yellow outline, respectively. The density maps are fitted with the in vitro structure of the GspD_β–GspS complex (PDB: 5ZDH). The secretin, N3, N2, N1, and N0 domains are marked with dashed lines. **h** Immunoblotting results of GspD_β and GspS overexpression by using anti-His antibodies. BL21 (DE3) cells (WT) and BW25113-*ΔgspS* cells (*ΔgspS*) without plasmid transformation were used as the control. **i** Detection of the membrane location of GspD_β through sucrose density gradient centrifugation followed by western blot analysis using anti-His tag antibodies (the bacteria strains are marked on the left, and all the bands correspond to GspD_β). The expression and membrane location experiments of GspD_β were repeated three times independently with similar results. IM inner membrane, OM outer membrane, WT wild type. Source data are provided as a Source Data file.

observed may not represent the exact scenario happening in an unmanipulated bacterium cell.

We observed GspD_β multimer structures on both the bacterial inner and outer membranes. We show that: (1) when *gspS* is knocked out in *E. coli*, GspD_β forms multimers on the inner membrane (Fig. 3f, g)[4]; (2) when *gspS* is not knocked out or overexpressed in *E. coli*, a small number of GspD_β multimers are still found on the inner membrane (Fig. 3d); (3) the undisturbed normal peptidoglycan pore size does not allow the passage of GspD_β multimers[30]; (4) when GspS exists, either from vector overexpression or endogenous genome expression, GspD_β

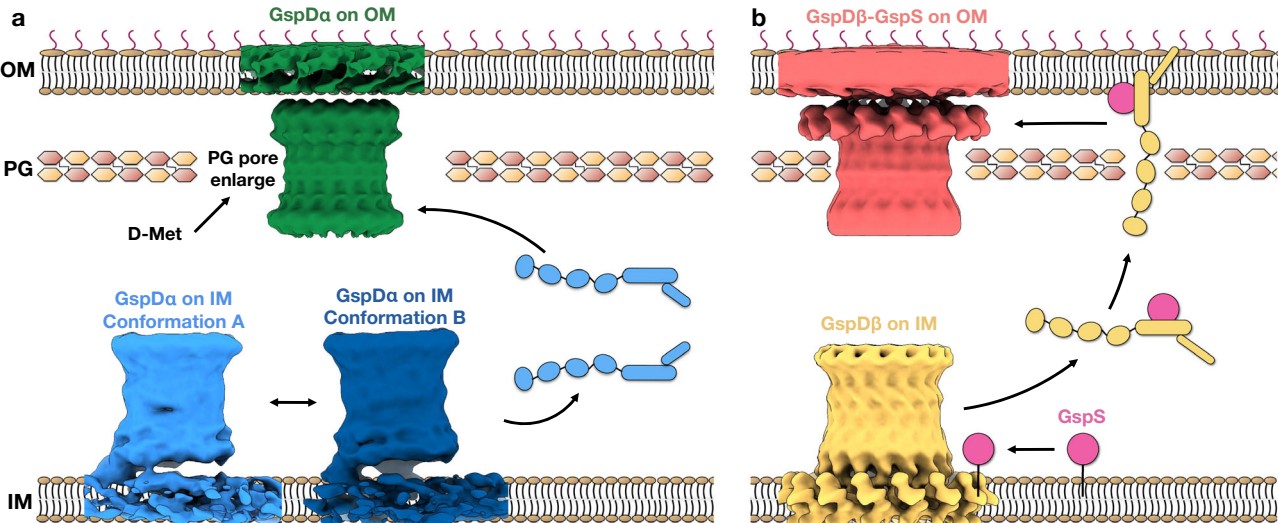

**Fig. 4 | Proposed models of the outer membrane translocation process of the GspD$_\alpha$ secretin (a) and the GspD$_\beta$ secretin (b). a** Firstly, GspD$_\alpha$ forms multimers on the inner membrane, and adopts an unstable and swinging conformation, changing between conformation A (light blue) and conformation B (dark blue). The multimer could dissociate from the inner membrane and enter the periplasm (light blue cartoon model), but it could not pass through the peptidoglycan and insert into the outer membrane. When the peptidoglycan pore size is enlarged by D-methionine, the multimer translocates to the outer membrane, where it exists in a more stable conformation (green). **b** GspD$_\beta$ first forms multimers on the inner membrane (yellow); then the pilotin GspS (pink) approaches and binds the GspD$_\beta$ monomer, transporting it to the outer membrane. On the outer membrane, GspD$_\beta$ assembles into a multimer again and GspS stays associated, forming a GspD$_\beta$–GspS multimer complex (light red). IM inner membrane, PG peptidoglycan, OM outer membrane, D-Met D-methionine.

multimers are found on the outer membrane (Fig. 3a, c), even if the peptidoglycan is undisturbed; (5) GspS binds to GspD$_\beta$ monomer in a 1:1 ratio[8]; (6) GspS itself is a lipoprotein that could translocate from the inner to the outer membrane through Lol pathway[15]; (7) we do not observe disassociated GspD$_\beta$ multimers in the periplasm; (8) GspD$_\beta$ multimers on the outer membrane exist in a complex with GspS, that is, GspS does not dissociate after transportation of GspD$_\beta$ to the outer membrane (Fig. 3b, e). Based on what we observed, we would propose a translocation model that, first GspD$_\beta$ self-assembles into multimers on the inner membrane, then GspS on the inner membrane binds to GspD$_\beta$, which dissociates into monomers. GspS then transports GspD$_\beta$ to the outer membrane, where GspD$_\beta$ assembles into multimers again with GspS associated, forming a GspD$_\beta$–GspS multimer complex (Fig. 4b). Because of the limitations of cryoET observations, we did not directly visualize the process of GspD$_\beta$ multimers on the inner membrane dissociating into monomers and entering the periplasm with GspS bound. Further investigations are needed to provide evidence for this process.

In our structures, the transmembrane region of GspD$_\alpha$ is only buried inside one leaflet of the membrane, instead of penetrating the membrane. This transmembrane pattern is unexpected and would be biophysically unreasonable but can be clearly observed from our cryo-ET structures. Notably, the in situ structure of the whole T2SS has been visualized[11], and it was indicated that the secretin of *Legionella pneumophila*, a *Klebsiella*-type secretin based on the secretin phylogenetic analysis[4], also punctures through only one leaflet of the outer membrane. It has been suggested that the transmembrane region of the secretin may have a more extended conformation in vivo based on these findings. Further investigations are necessary in order to explain why this membrane interaction could exist, and what is the in vivo conformation that makes this kind of transmembrane pattern possible.

Comparing the inner and outer membrane-located structures of GspD$_\alpha$ and GspD$_\beta$, we observe that GspD$_\alpha$ only transits one leaflet of the lipid bilayer, but GspD$_\beta$ transits through both leaflets (Supplementary Fig. 8). This difference could give GspD$_\beta$ higher efficiency when transporting substrates. On the *E. coli* genome, there are two T2SS gene operons, T2SS$_\alpha$ and T2SS$_\beta$. During evolution, T2SS$_\beta$ may be generated by gene duplication of the T2SS$_\alpha$ operon. However, T2SS$_\alpha$ is endogenously silenced, while T2SS$_\beta$ is actively expressed and functional[15]. Our result could provide a possible explanation, that is, the possibly higher substrate transport efficiency gives GspD$_\beta$ an advantage during evolution, making it the preferred and expressed T2SS in many bacteria.

In summary, we determined four in situ structures of the T2SS secretin: both the inner and outer membrane structures of GspD$_\alpha$ and GspD$_\beta$. GspD$_\alpha$ exists in an unstable form on the inner membrane, but forms firm and stable connections with the outer membrane. The GspD$_\beta$ multimer exists on the inner membrane in wild-type *E. coli*, and GspD$_\beta$ on the outer membrane exists together with GspS. Taken together, these results add to the knowledge of the membrane interactions of the T2SS secretins and their outer membrane targeting process, providing a comprehensive model for the secretin biogenesis process of *Proteobacteria*.

## Methods

### Bacterial strains, plasmids, chemicals, and protein expression

*E. coli* BW25113-Δ*gspS* strain was obtained from the Nara Institute of Science and Technology in Japan, from the Keio collection[31,32]. *E. coli* BL21 (DE3)-Δ*gspS* strain was constructed by Ubigene Biosciences Co., Ltd. (Guangzhou, China). The DNA sequences encoding GspD$_\alpha$ (*E. coli* K12 strain), GspD$_\beta$ (ETEC H10407 strain), and GspS (ETEC H10407 strain) were synthesized by General Biosystems Co., Ltd. (Anhui, China). D-methionine was purchased from Sigma (catalog number: M9375-5G).

The *gspD$_\alpha$* from the *E. coli* K12 strain with no tag or with a C-terminal hexahistidine tag was cloned into pETDuet-1, yielding pETDuet-*gspD$_\alpha$* and pETDuet-*gspD$_\alpha$*-his, which were expressed in *E. coli* BL21 (DE3), respectively. The vector with no tag was used for cryo-ET imaging, and the vector with the hexahistidine tag was used to perform biochemistry tests. The *gspD$_\beta$* of ETEC H10407 strain with hexahistidine tag on the C terminal was inserted into pETDuet-1, yielding pETDuet-*gspD$_\beta$*-his, which was expressed in *E. coli* Rosetta (DE3) and used for cryo-ET imaging and expressed in *E. coli* BL21 (DE3) for biochemistry experiments. The gene of *gspS* was cloned into pETDuet-*gspD$_\beta$*-his with a hexahistidine tag at the C terminus, yielding

pETDuet-$gspD_\beta$-his-$gspS$-his, which was expressed in *E. coli* BL21 (DE3) and used for the biochemistry experiments. The $gspD_\beta$ with hexahistidine tag and $gspS$ with no tag from ETEC H10407 strain were both inserted into pETDuet-1's two multi-cloning sites, yielding pETDuet-$gspD_\beta$-his-$gspS$, which was expressed in *E. coli* BL21 (DE3) and used for cryo-ET imaging. The $gspD_\beta$ of ETEC H10407 strain with hexahistidine tag was inserted into pBAD, yielding pBAD-$gspD_\beta$-his, which was expressed in BW25113-$\Delta gspS$ strain and used for cryo-ET imaging, and was expressed in BL21 (DE3)-$\Delta gspS$ strain and used for biochemistry experiments. For cryo-ET imaging experiments, we obtained minicells by co-transforming the *E. coli* strains with pBS58, which increases the frequency of cell division by constitutively expressing cell division-related genes[33]. The obtained minicell thickness ranges from 130 to 250 nm. *E. coli* cells only transformed with pBS58 and not with any protein expression vectors were used as negative control cells.

Cells were grown at 37 °C in LB medium supplemented with appropriate antibiotics (final concentrations of 100 µg/ml ampicillin and/or 50 µg/ml kanamycin), and 40 mM D-methionine when needed. Protein expression was induced at an $OD_{600}$ of 0.8, by adding 0.5 mM IPTG or 0.02% arabinose at 20 °C overnight. Protein expression was examined by western blotting analysis using anti-His tag antibodies. For experiments with $GspD_\alpha$, the anti-His tag antibodies were diluted 10,000-fold. For experiments with $GspD_\beta$, the anti-His tag antibodies were diluted 1000-fold. The antibodies for the control EF-Tu were from previous publications[34,35] and were diluted 20,000-fold.

### Sample preparation

To perform minicell separation, *E. coli* cultures were first centrifuged at 1000 × *g* for 40 min to precipitate large cells. The supernatant was taken and centrifuged at 20,000 × *g*, for 10 min. The precipitation was washed with PBS buffer for one time, then resuspended to an $OD_{600}$ of 10. Cells were mixed with 6 nm BSA fiducial gold (Aurion) and then, 3 µl mixture was deposited onto freshly glow-discharged, continuous carbon film-covered grids (Quantifoil Cu R3.5/1 with 2 nm continuous carbon film, 200 mesh). Using Vitrobot Mark IV (FEI), the grids were blotted with filter papers for 4 s and plunged into liquid ethane. Samples were stored in liquid nitrogen.

### Cryo-ET data collection

All tilt series were collected using 5° angular step, from −50° to +50°, defocus from −1.5 to −5 µm, and a total dose of about 100 e−/Å². For the $GspD_\alpha$ on the inner and outer membrane dataset, the sample was imaged using the bidirectional data collection scheme (starting angle −30°) on a 300 kV FEI Titan Krios microscope with a Gatan K2 Summit direct electron detector camera, using the SerialEM software. The magnification was 81,000×, with a pixel size of 1.76 Å. For the other datasets and the control dataset, the sample was imaged using a dose symmetric data collection scheme on a 200 kV FEI Glacios microscope with a Falcon 4 direct electron detector camera, using the Tomography software. The magnification was 92,000×, with a calibrated pixel size of 1.52 Å (Supplementary Table 1).

### Cryo-ET data processing of $GspD_\alpha$ on the inner membrane dataset

**EMAN2 refinement pipeline.** The raw frames were aligned by MotionCorr2[36]. The tilt series alignment, tomogram reconstruction, and CTF correction were performed in EMAN2[26]. For the refinement, 32,915 particles were manually picked from 250 tilt series and extracted using a box size of 256. After generating an initial model with a small set of particles that have a good signal-to-noise ratio, 4 iterations of subtomogram refinement were performed. The worst 20% of particles were excluded based on their similarity to the averaged structure.

**Using the cell membrane feature to assist particle alignment.** Due to the shape of $GspD_\alpha$, its top view and bottom view are hard to distinguish, and after the refinement, many top-view particles were wrongly aligned by about 180°. To correct this, an algorithm was developed that decides a center for each tomogram based on all the particles picked in this tomogram and draws a vector from the particle position to the center (or the opposite direction if the parameter "invert" is toggled). If a particle's orientation from the refinement result is not facing the same side as its corresponding vector, its orientation will be rotated 180°. After the correction, a new particle orientation file was written and used as input for the next iteration of refinement. In the next iterations, local refinement was used to make sure that the particle orientation will not rotate back to the wrong side. And three more iterations of refinement were performed to achieve the final structure.

**Symmetry determination and particle radius measurement.** To determine the particle symmetry, we first erected each particle based on its orientation from the refinement and projected them to 2D images, and then performed 2D classification in EMAN2, but the result did not show distinctly different classes or explicit symmetry. We also did refinement with C12, C14, and C16 symmetry, but the resulting density map did not show any obvious feature differences or feature improvements compared to the C15 structure. To measure the particle radius, we applied a low pass filter to the particle 2D projection image, calculated the mean radial intensity along the radial axis, and then recorded the index corresponding to the maximum intensity value (Supplementary Fig. 2). The particle radius is pixel number times angstrom per pixel.

**Symmetry release and particle movement trajectory calculation.** To visualize the membrane connecting region of $GspD_\alpha$ more clearly, we did a focused classification, using a mask that is focusing on the membrane connecting region of the density map (Supplementary Fig. 3, dashed line box). Within the results, there is one class that shows C1 features (Supplementary Fig. 3, red line box). To achieve this C1 structure, we did a symmetry release using the whole dataset: with the orientation of the symmetry axis unaltered, each particle is allowed to rotate around the symmetry axis and search for a best-fitted symmetry unit for one iteration, followed by averaging of subtomograms. Then, the particle orientations are subject to two iterations of gold standard refinement, doing only a local search.

We have observed tilted inserted particles from the tomograms (Fig. 1e−g), which indicates that, for these particles in the alignment process, if we give a reference map where the membrane and the particle symmetry axis are perpendicular, the membrane density and the protein density could not be aligned correctly at the same time. If the membrane density plane is correctly aligned, the protein density orientation will be inaccurate. To resolve this problem, we did a focused refinement. A mask was generated enclosing the non-transmembrane part of the protein, excluding the membrane density, and one iteration of alignment was performed using the masked structure as a reference (Supplementary Fig. 3g). In this iteration, the protein density part should be aligned correctly. We then compared the orientation from the focused refinement with that from the overall refinement, and the orientation difference represents the protein tilt angle. The orientation difference for each particle could be plotted, and a trajectory was calculated. To show the movement, the full trajectory was sectioned into several intervals, and class averages for each interval were calculated representing the corresponding conformation (Supplementary Fig. 3h).

### Cryo-ET data processing of the other datasets

The data processing workflow for the $GspD_\alpha$ on the outer membrane dataset, $GspD_\beta$−GspS complex dataset, and $GspD_\beta$ on the inner membrane dataset resembles the $GspD_\alpha$ on the inner membrane dataset, but simpler. After motion correction and tomogram reconstruction,

particles were picked from tomograms. After generating the initial model, the refinement was firstly done with C15 symmetry for 4 iterations, and then particle orientations were corrected using the cell membrane geometry. Then another 4 iterations of local refinement were done to achieve the final structure (for C5 refinements, we only changed the symmetry input to C5, and other parameters were the same). The symmetry release protocol of the $GspD_\alpha$ on the outer membrane dataset followed that of the $GspD_\alpha$ on the inner membrane dataset.

## Extraction of outer membrane protein

Outer membrane proteins were extracted using a bacterial membrane protein extraction kit (BestBio). Briefly, *E. coli* cells expressing the protein of interest were harvested by centrifugation, washed twice with PBS, and suspended in Extracting Solution A (containing protease inhibitor mixture). After 1 h of shaking at 4 °C, the suspension was centrifuged at $12,000 \times g$ for 5 min at 4 °C to remove the pellet. The supernatant was taken and incubated at 37 °C for 1 h, after which it could be observed that the liquid was divided into two layers. The liquid at the bottom was collected as the outer membrane proteins.

## Membrane separation of the *E. coli* cell envelope

Separation of inner and outer membranes by isopycnic sucrose density gradient centrifugation was performed as described previously[37]. Briefly, *E. coli* cells expressing the protein of interest were harvested by centrifugation and suspended in buffer A (10 mM Tris-HCl, pH 7.5; 0.5 M sucrose; 10 mg/ml lysozyme; 1.5 mM EDTA). After incubation on ice for 7 min, the suspension was centrifugated at $10,000 \times g$ for 10 min at 4 °C, and then the pellet was resuspended in buffer B (10 mM Tris-HCl, pH 7.5; 0.2 M sucrose; 1 M MgCl$_2$) containing RNase/DNase nuclease reagent and protease inhibitor cocktail. The mixture was lysed by sonication and centrifuged at $6169 \times g$ for 10 min at 4 °C to remove the cell debris. Subsequently, the supernatant was pelleted by ultracentrifugation at $184,500 \times g$ for 1 h at 4 °C and then suspended in buffer C (1 mM Tris-HCl, pH 7.5; 1 mM EDTA,) containing 20% sucrose to get the total membrane fraction. After that, buffer C containing 73% sucrose and buffer C containing 53% sucrose were layered in a 13 ml tube (Beckman Coulter), starting from the bottom, and finally, 0.5 ml of the total membrane fraction was layered on the top, followed by buffer C containing 20% sucrose to fill up the 13 ml tube. Then, the density gradients were centrifuged at $288,000 \times g$ for 16 h at 4 °C, and the tawny band at the interface between 20% and 53% sucrose layers and the white band at the interface between 53% and 73% sucrose layers were collected as inner membrane fraction and outer membrane fraction, respectively.

## Reporting summary

Further information on research design is available in the Nature Portfolio Reporting Summary linked to this article.

## Data availability

Density maps are deposited in EMDB with accession codes: EMD-29702 ($GspD_\alpha$ on the inner membrane), EMD-29703 ($GspD_\alpha$ on the outer membrane), EMD-29698 ($GspD_\beta$ on the inner membrane), EMD-29697 ($GspD_\beta$–GspS on the outer membrane, expressing only $GspD_\beta$), and EMD-29696 ($GspD_\beta$–GspS on the outer membrane, expressing $GspD_\beta$ and GspS). Source data are provided in this paper.

## Code availability

All the relevant codes are available in the EMAN2 package.

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

## Acknowledgements

This work was supported by R01GM143380 and R01HL162842, the Welch Foundation Q-2173-20230405, and BCM BMB department seed funds to Z.W.; the National Natural Science Foundation of China (No. 82072312), and the Natural Science Foundation of Jiangsu Province (No. BK20211053) to X.S.; Postgraduate Research & Practice Innovation Program of Jiangsu Province (No. KYCX22_2925); and NIH R01GM080139 to S.J.L. CryoEM data was collected at the Baylor College of Medicine CryoEM ATC and UTHealth cryoEM core, which includes equipment purchased under the support of CPRIT Core Facility Award RP190602. We thank Dr. Jun Liu and Dr. Bo Hu for sharing the pBS58 plasmid. We thank Hongjiang Wu for discussions about vector design and biochemistry experiments, Valerie Dalton for revising the paper language, and Snekalatha Raveendran for data backup.

## Author contributions

Z.W., X.S., and Z.Y. designed experiments. Z.Y. performed sample freezing and cryo-ET imaging. Z.Y., M.C., and T.H. did cryo-ET data processing. Y.W. and W.Z. performed biochemistry experiments. Z.Y. wrote the initial paper. Z.W., X.S., S.J.L., Y.W., and M.C. revised the paper.

## Competing interests

The authors declare no competing interests.
