## [Peer Review File · Nature Communications]

Membrane translocation process revealed by in situ structures of type II secretion system secretinsREVIEWER COMMENTS

Reviewer #1 (Remarks to the Author):

This manuscript describes in situ structures of the GspDa and GspDb secretins on the inner and outer membranes of *E. coli*. The OM structure of GspDb also reveals the density for the GspS chaperone. While the high resolution structures of these proteins have been defined previously in vitro, the results of this study are of high significance- both in advancing the understanding of type two secretion-system (T2SS) biogenesis and in the advancement of technical methodologies. Important biological observations include the presence of GspDa/b oligomers on the inner membrane, but with membrane interactions that are clearly less stable than their final destination on the outer membrane, the unique C15 symmetry (as opposed to the mixed symmetries in vitro), and the location of the GspS chaperone on the OM structures of GspDb. Technical advances include the ability to incorporate the membrane location into the orientation for single particle averaging,

Overall, I found this to be a high quality body of work. I have only a few minor suggestions to improve clarity:

Typo p.4, line 76: Change secret to secrete.

Typo p.4, line 77: Change secret to secrete.

p.4, line 81: Change complex to complexes

In pairing text on page 5 with Figure 1, I was not sure what the authors meant when they said 'parentheses object'. (line 100) Can this be specifically labelled in a figure?

Page 6, line 110. The histogram would be more compelling if authors state the expected radius of a C12 and C15 structure so that readers can interpret for themselves.

Page 6, line 117. Define "D" symmetric features to improve clarity.

Page 7, line 143. While true that the strength of the interaction would almost certainly change due to avidity affects, might be more accurate to say that the number of connections is different.

Page 14, line 310. This is the first mention of them as T2SSa and T2SSb and not explicit that one has GspDa and one had GspDb. Better to define this in the introduction on page 4 at line 76.

Language around use of D-methionine. In reviewing, reference 26, the mechanism of how D-methionine affects the cell wall is unclear. While it is possible that D-methionine increases pore size, it is also possible that D-methionine affects the rate of peptidoglycan remodeling. In this case, it was an effective tool for visualizing OM-tethered complexes, but for clarity, it would be good to indicate to indicate that increased pore size is a possible mechanism, but that other possibilities exist.

Reviewer #2 (Remarks to the Author):

In situ structures of secretins from bacterial type II secretion system reveal their membrane interactions and translocation process

Many bacteria utilize the type II secretion system (T2SS) to secrete substrates and/or to cause pathogenesis. A major component of the T2SS machinery is the large barrel-shaped outer membrane complex, known as the secretin complex. In this manuscript, the authors investigated in situ architectures of two phylogenetically distinct families of secretins and proposed a mechanism for the outer membrane biogenesis of the secretin complex. The manuscript is clearly written and provides high-res in situ structure of the secretin complex. In addition, the authors developed a couple of useful tools in the EMAN2 workflow that the cryoET community might find useful. However, in situ structure of the complete *Legionella pneumophila* T2SS was already reported a few years ago by the Jensen lab (Nat Micro, 2019; surprisingly, not cited/discussed here) and there are several near-atomic resolution structures of the secretin complex published by many other groups (Chernyatina et al, Nat Comms, 2019; Yan et al, Nat Micro. 2017; Hay et al, J Bac 2018 etc.). Therefore, this manuscript does not provide any significant knowledge advancement. The proposed mechanism is not fully convincing (see below). There are additional major and minor concerns, as listed below.

Major:

1. The author imaged a highly overexpressed (artificial) system where each cell is expressing ~132 (33k/250 or more) T2SS secretins. While I am not certain about *E. coli*, some other gram-negative bacteria, on average, express less than one T2SS per cell. While obtaining an in situ structure of the secretin complex from such a system is convenient (just like the efflux pump), proposing a mechanism of biogenesis based on this system is unreliable. There are many groups that have imaged a variety of bacterial cells using cryoET and to my knowledge, no one has ever seen T2SS secretin attached to the IM in an inverted way.

2. If we assume that the secretin is initially indeed attached to the IM in an inverted way, what happens to the other cytoplasmic/IM-associated proteins? Are they not localized at this point? After the authors grew cells in the presence of 40 mM D-methionine, why there are still more secretins associated with the IM than OM (Fig. 1h,i)? If the IM association is an unstable intermediate step towards biogenesis, I would expect more of the complexes associated with the OM, particularly after cells are fed with 40 mM D-methionine.

3. The authors should image native T2SSs (not just the secretin) without targeting factors and show that the secretins are localized to the IM and inverted. This might bring some traction to the proposed mechanism. But as mentioned above, to date no one has seen T2SS secretin attached to the IM in any bacterial cellular tomograms.

4. Since the compositions of the IM and OM are different, an association of the secretin complex will likely differ in these two membranes. Therefore, unstable association with the IM is not enough of evidence to call this an intermediate step.

Minor:

1. Line 56: How the T2SS secretin interacts with the membrane has already been visualized. The authors should go through the PMID: 31754273 (extended data fig. 3). Some of the conclusions of the current MS are already reported before. The authors should reconcile published work.

2. Line 65: There are T2SSs without pilotin or GspA/B, therefore likely more pathways exist.

3. Line 107: After a series of high-res structures of the secretin complex (Chernyatina et al, *Nat Comms*, 2019; Yan et al, *Nat Micro*. 2017; Hay et al, *J Bac* 2018), we already know that the T2SS secretin is 15-fold symmetric. Ref 24/25 that mention 12-fold symmetry are too old.

4. Line 193: Needs more clarification on why tested C5 symmetry. Might be hard for people outside the structural biology field to follow.

5. Supplementary Movie 1, frame ~7 sec: side and top both views of secretins are visible. Does this mean cells were unhealthy/sick?

Reviewer #1 (Remarks to the Author):

This manuscript describes *in situ* structures of the GspDa and GspDb secretins on the inner and outer membranes of *E. coli*. The OM structure of GspDb also reveals the density for the GspS chaperone. While the high resolution structures of these proteins have been defined previously *in vitro*, the results of this study are of high significance- both in advancing the understanding of type two secretion-system (T2SS) biogenesis and in the advancement of technical methodologies. Important biological observations include the presence of GspDa/b oligomers on the inner membrane, but with membrane interactions that are clearly less stable than their final destination on the outer membrane, the unique C15 symmetry (as opposed to the mixed symmetries *in vitro*), and the location of the GspS chaperone on the OM structures of GspDb. Technical advances include the ability to incorporate the membrane location into the orientation for single particle averaging,

Overall, I found this to be a high quality body of work. I have only a few minor suggestions to improve clarity:

Typo p.4, line 76: Change secret to secrete.

Thank you for the suggestion. We have changed the text accordingly (line 79).

Typo p.4, line 77: Change secret to secrete.

We have changed the text accordingly (line 81).

p.4, line 81: Change complex to complexes

We have changed the text accordingly (line 84).

In pairing text on page 5 with Figure 1, I was not sure what the authors meant when they said 'parentheses object'. (line 100) Can this be specifically labelled in a figure?

We would like to offer a direct impression to the reader that the protein structure is similar to a hollow cylinder, and its cross-section will have different shapes from different views. The top view will appear as a circle, its side view will appear as two paired lines in the projected image,

while the tilted view will be two paired curved lines. We realize that this description may be misleading and confusing. The corresponding text has been changed to: “we identified particles in the top view and side view as in circles and a pair of curved lines attaching to the inner membrane, respectively” (lines 102-103).

Page 6, line 110. The histogram would be more compelling if authors state the expected radius of a C12 and C15 structure so that readers can interpret for themselves.

We have modified Supplementary Fig. 2 and its legends accordingly. In Supplementary Fig. 2, a red dashed line and a blue dashed line were added to indicate the expected positions of the C12 and C15 particle radii, respectively.

Page 6, line 117. Define “D” symmetric features to improve clarity.

Thank you for the suggestion. We have modified the text (line 121).

Page 7, line 143. While true that the strength of the interaction would almost certainly change due to avidity affects, might be more accurate to say that the number of connections is different.

Thank you for the suggestion. However, the connection strength between two densities is not a quantitative/integral measurement. To avoid confusion, we changed the phrasing in this sentence (lines 147-148).

Page 14, line 310. This is the first mention of them as T2SSa and T2SSb and not explicit that one has GspDa and one had GspDb. Better to define this in the introduction on page 4 at line 76.

We have modified the text in the introduction accordingly (lines 78-81).

Language around use of D-methionine. In reviewing, reference 26, the mechanism of how D-methionine affects the cell wall is unclear. While it is possible that D-methionine increases pore size, it is also possible that D-methionine affects the rate of peptidoglycan remodeling. In this case, it was an effective tool for visualizing OM-tethered complexes, but for clarity, it would be good to indicate that increased pore size is a possible mechanism, but that other possibilities exist.

Thank you for the suggestion. We realize that in reference 26, it is both possible that D-methionine incorporates into the muropeptides, and that D-methionine directly influences the

synthesis of peptidoglycan and therefore peptidoglycan remodeling. However, we think that both mechanisms will eventually result in enlarged pore size and decreased crosslinking, even if momentarily, so that GspD_α multimer could traverse the peptidoglycan layer. We have modified the text (lines 180-183) for clarity.

Reviewer #2 (Remarks to the Author):

In situ structures of secretins from bacterial type II secretion system reveal their membrane interactions and translocation process

Many bacteria utilize the type II secretion system (T2SS) to secrete substrates and/or to cause pathogenesis. A major component of the T2SS machinery is the large barrel-shaped outer membrane complex, known as the secretin complex. In this manuscript, the authors investigated in situ architectures of two phylogenetically distinct families of secretins and proposed a mechanism for the outer membrane biogenesis of the secretin complex. The manuscript is clearly written and provides high-res in situ structure of the secretin complex. In addition, the authors developed a couple of useful tools in the EMAN2 workflow that the cryoET community might find useful. However, in situ structure of the complete *Legionella pneumophila* T2SS was already reported a few years ago by the Jensen lab (Nat Micro, 2019; surprisingly, not cited/discussed here) and there are several near-atomic resolution structures of the secretin complex published by many other groups (Chernyatina et al, Nat Comms, 2019; Yan et al, Nat Micro. 2017; Hay et al, J Bac 2018 etc.). Therefore, this manuscript does not provide any significant knowledge advancement. The proposed mechanism is not fully convincing (see below). There are additional major and minor concerns, as listed below.

We thank the reviewer for pointing out the missing references, and we have now added the citation and included more discussion about the relationship between this work and previous literature. Specifically, we would argue that by studying the secretin inside living cells and determining their structures at subnanometer resolution in native membrane systems, we, in fact, brought knowledge advancement to the structure and biogenesis process of the T2SS. Importantly, by imaging the secretin in an overexpressed system *in situ*, we revealed a novel pathway of the secretin membrane translocation process.

Notably, in addition to Ghosal et al, Nat Micro, 2019, we have verified the symmetry of secretin *in vivo*, and achieved a higher resolution. Additionally, we visualized both the *Klebsiella*-type secretin GspD α and the *Vibrio*-type secretin GspD β *in vivo*, in both the IM and OM, while Ghosal et al, Nat Micro, 2019 visualized the secretin of *Legionella pneumophila*, a *Klebsiella*-type secretin (taxonomy based on PMID: 23326233, Table S2) on the OM, together with the full T2SS complex. About the transmembrane region, there is a difference between our observations and Ghosal et al, Nat Micro, 2019. PMID: 31754273 (extended data fig. 3) suggested that both *Klebsiella*-type and *Vibrio*-type secretins transit through one leaflet of the OM. However, we are suggesting that GspD α (*Klebsiella*-type) transits through one leaflet, while GspD β (*Vibrio*-type) transits through two leaflets of the OM.

Our data generated a new hypothesis about the biogenesis process of the secretin. However, confirming it at native conditions with native concentrations of secretin in bacteria cells (as the reviewer mentioned, less than one particle per cell) with cryoET is unfortunately infeasible. Our studies using an expression system could help us to capture possible biological processes that could happen in a living bacteria cell context with cryoET. Our novel observations in the cell could help the secretin research community to generate and test new hypotheses in the secretin membrane translocation process in Gram-negative bacteria. We have modified the introduction to reconcile with previous literature (lines 56-61) and added a comparison in the discussion (lines 318-324). The abstract was also modified to specify that we are proposing hypotheses (line 27).

Major:

1. The author imaged a highly overexpressed (artificial) system where each cell is expressing ~132 (33k/250 or more) T2SS secretins. While I am not certain about *E. coli*, some other gram-negative bacteria, on average, express less than one T2SS per cell. While obtaining an in situ structure of the secretin complex from such a system is convenient (just like the efflux pump), proposing a mechanism of biogenesis based on this system is unreliable. There are many groups that have imaged a variety of bacterial cells using cryoET and to my knowledge, no one has ever seen T2SS secretin attached to the IM in an inverted way.

In any structure biology study, we always need to seek the balance between the nativeness of the system and the level of detail we can observe. In single particle analysis of purified secretins, the complex is completely taken out of its native environment, and it is impossible to study the

protein-membrane interaction from the high-resolution structure. On the other hand, while it is possible to image the entire T2SS in native cellular systems using cryoET, it is extremely challenging to obtain a large enough dataset due to the low expression level of the system. This greatly limits the resolution we can achieve, and the confidence in any biological conclusion we can make from the data. Therefore, we decided to use the current overexpression system to gain more insight into the secretin membrane translocation process. While it is certainly more artificial than the native structure of T2SS on cell surfaces, it is much more biologically relevant than the purified systems, and can still produce enough particles to solve the structures at a high enough resolution to study the protein-membrane interaction and visualize the transient states of the translocation process. In sum, we believe our result fills a critical gap in the understanding of T2SS and provides a bridge between the high-resolution structure studies and the more qualitative *in situ* biochemistry experiments.

As the reviewer pointed out, the endogenous expression level is less than one T2SS per cell in many other bacteria. In native cells, the chance is extremely low that one can capture secretin inverted on the IM, and it will be challenging to investigate the biogenesis process of the secretin in this scheme. Therefore, the induced expression is required to visualize and achieve high-resolution *in situ* structures for GspD α . Here in our study, we aim to propose the biogenesis mechanism based on our observations. We also admit that in the real native condition, the whole biogenesis process may be more complicated than the situation we mimicked here in the *E. coli*. Due to the rarity of T2SS in native *E. coli*, we cannot visualize it using cryoET, but our result is plausible and worth further investigation by the community.

2. If we assume that the secretin is initially indeed attached to the IM in an inverted way, what happens to the other cytoplasmic/IM-associated proteins? Are they not localized at this point?

That is correct. We would not anticipate the other T2SS components to localize and associate with the inverted secretin on the IM. We did not express those other components in this experiment, so we cannot conclusively state this, but we see no plausible biological reason that the other components would be inverted as well.

After the authors grew cells in the presence of 40 mM D-methionine, why there are still more secretins associated with the IM than OM (Fig. 1h,i)? If the IM association is an unstable

intermediate step towards biogenesis, I would expect more of the complexes associated with the OM, particularly after cells are fed with 40 mM D-methionine.

It is possible that other mechanisms exist to help GspD α relocate to the OM. D-methionine is not the native mechanism for secretin translocation, so we do not expect it to be sufficient to relocate all particles to the OM. It simply demonstrates that translocation is possible when the peptidoglycan pore is enlarged. Actually, we also tried the method of expressing GspAB together with GspD α , following the rationale in lines 67-68. In this dataset, we also observed particles on the OM, but the number/ratio of particles on the OM is lower than that using D-methionine, so this data was not included in the manuscript. Therefore, in the native system, it is likely that other unknown components are responsible for the process. We label this state as “unstable” since the IM interaction is less stable compared to the OM interaction. To clarify, we have added this content to the discussion (lines 290-293). We appreciate your assistance in refining our statements to strengthen our conclusions.

3. The authors should image native T2SSs (not just the secretin) without targeting factors and show that the secretins are localized to the IM and inverted. This might bring some traction to the proposed mechanism. But as mentioned above, to date no one has seen T2SS secretin attached to the IM in any bacterial cellular tomograms.

We agree with the reviewer and would love to be able to do this experiment, but the event is simply too rare for this technique. We believe by making our hypothesis, investigators using techniques outside our expertise may be inspired to test this observation.

4. Since the compositions of the IM and OM are different, an association of the secretin complex will likely differ in these two membranes. Therefore, unstable association with the IM is not enough of evidence to call this an intermediate step.

We agree that different compositions of two membranes could cause different interactions. However, the unstable association is not the only observation we based on to call it an intermediate step. We also combined our observations that, without D-methionine, all the GspD α particles are on the IM, but when adding D-methionine, GspD α could relocate to the OM. Therefore, we infer that the IM located GspD α is an intermediate step.

Minor:

1. Line 56: How the T2SS secretin interacts with the membrane has already been visualized. The authors should go through the PMID: 31754273 (extended data fig. 3). Some of the conclusions of the current MS are already reported before. The authors should reconcile published work.

Thank you for the suggestion. We have modified the text accordingly (lines 56-61).

2. Line 65: There are T2SSs without pilotin or GspA/B, therefore likely more pathways exist. We agree that more pathways possibly exist. The text has been modified (lines 71-72).

3. Line 107: After a series of high-res structures of the secretin complex (Chernyatina et al, Nat Comms, 2019; Yan et al, Nat Micro. 2017; Hay et al, J Bac 2018), we already know that the T2SS secretin is 15-fold symmetric. Ref 24/25 that mention 12-fold symmetry are too old.

We agree that the protein has been verified to have C15 symmetry *in vitro*. However, one of our research highlights is that we verified that the secretin has C15 symmetry *in vivo* within a membrane environment. To avoid complexity, we have modified the text (lines 109-111).

4. Line 193: Needs more clarification on why tested C5 symmetry. Might be hard for people outside the structural biology field to follow.

Thank you for the suggestion. We have modified the text (lines 200-201).

5. Supplementary Movie 1, frame ~7 sec: side and top both views of secretins are visible. Does this mean cells were unhealthy/sick?

This is primarily an artifact of the reconstruction geometry. The cell orientation is tilted with respect to the x-y slices in the movie. Consequently, side views and top views can be observed simultaneously, without the cells being unhealthy.

REVIEWERS' COMMENTS

Reviewer #1 (Remarks to the Author):

The authors have answered/addressed the points raised in my prior review.

Reviewer #2:

While the structural work is fantastic, I am still not very convinced that the proposed mechanism is correct. If the authors acquire 100-150 tomograms of cells with endogenous T2SSs, there is a good chance that the authors will see at least 250-300 particles. If they see 1/2 particles associated with the IM, that would support their mechanism. If not, then how can we conclude that the IM association is not an artifact of over-expression. Personally, I have collected many tomograms of Legionella cells and observed ~400+ T2SS particles and never seen a single secretin associated with the IM.

I am not saying that the authors should do an average of the endogenous T2SS. My point is can they see a 'single' T2SS secretin associated with the IM in native expression system!

In the end if this manuscript is accepted, the authors should at the least highlight that the IM association could also be an artifact of over-expression and tone down their claims.

Reviewer #1:

The authors have answered/addressed the points raised in my prior review.

We thank the reviewer for the comment.

Reviewer #2:

While the structural work is fantastic, I am still not very convinced that the proposed mechanism is correct. If the authors acquire 100-150 tomograms of cells with endogenous T2SSs, there is a good chance that the authors will see at least 250-300 particles. If they see 1/2 particles associated with the IM, that would support their mechanism. If not, then how can we conclude that the IM association is not an artifact of over-expression. Personally, I have collected many tomograms of Legionella cells and observed ~400+ T2SS particles and never seen a single secretin associated with the IM.

I am not saying that the authors should do an average of the endogenous T2SS. My point is can they see a 'single' T2SS secretin associated with the IM in native expression system!

In the end if this manuscript is accepted, the authors should at the least highlight that the IM association could also be an artifact of over-expression and tone down their claims.

We thank the reviewer for the suggestion. We have toned down our claims and added this content to the first paragraph of the Discussion part (lines 295-297).